# Lipid-Based Nanoparticles in Delivering Bioactive Compounds for Improving Therapeutic Efficacy

**DOI:** 10.3390/ph17030329

**Published:** 2024-03-01

**Authors:** Priya Patel, Kevinkumar Garala, Sudarshan Singh, Bhupendra G. Prajapati, Chuda Chittasupho

**Affiliations:** 1Department of Pharmaceutical Sciences, Saurashtra University, Rajkot 360005, Gujarat, India; 2School of Pharmaceutical Sciences, Atmiya University, Rajkot 360005, Gujarat, India; kevin.garala@atmiyauni.ac.in; 3Office of Research Administration, Chiang Mai University, Chiang Mai 50200, Thailand; sudarshansingh83@hotmail.com; 4Department of Pharmaceutical Sciences, Faculty of Pharmacy, Chiang Mai University, Chiang Mai 50200, Thailand; 5Shree S. K. Patel College of Pharmaceutical Education and Research, Ganpat University, Kherva 384012, Gujarat, India

**Keywords:** bioactive compound, drug delivery, nanotechnology, lipid-based nanoparticles, therapeutic efficacy

## Abstract

In recent years, due to their distinctive and adaptable therapeutic effects, many natural bioactive compounds have been commonly used to treat diseases. Their limited solubility, low bioavailability, inadequate gastrointestinal tract stability, high metabolic rate, and shorter duration of action limited their pharmaceutical applications. However, those can be improved using nanotechnology to create various drug delivery systems, including lipid-based nanoparticles, to adjust the compounds’ physicochemical properties and pharmacokinetic profile. Because of the enormous technical advancements made in the fundamental sciences and the physical and chemical manipulation of individual atoms and molecules, the subject of nanotechnology has experienced revolutionary growth. By fabricating certain functionalized particles, nanotechnology opens an innovative horizon in research and development for overcoming restrictions, including traditional medication administration systems. Nanotechnology-driven bioactive compounds are certain to have a high impact and clinical value for current and future uses. Lipid-based nanotechnologies were shown to deliver a range of naturally occurring bioactive compounds with decent entrapment potential and stability, a successfully controlled release, increased bioavailability, and intriguing therapeutic activity. This review outlines bioactive compounds such as paclitaxel, curcumin, rhodomyrtone, quercetin, kaempferol, resveratrol, epigallocatechin-3-gallate, silymarin, and oridonin, fortified within either a natural or synthetic lipid-based drug delivery system based on nanotechnology and their evaluation and clinical considerations.

## 1. Introduction

In recent years, fatty acids, essential oils, vitamins, polyphenols, flavonoids, terpenes, alkaloids, peptides, and proteins have been the most reported bioactive compounds [1]. These bioactive compounds offer a wide range of applications, for example, the ability to inhibit enzymes, scavenge free radicals, and prevent cell proliferation [2]. Polyphenols are secondary plant metabolites containing at least one aromatic ring bearing hydroxyl groups as structural components. Phenolics and flavonoids are identified due to their potent antioxidant, reactive oxygen species inhibition, and cell-protective properties. Several studies have suggested that the radical scavenging efficacy of tocopherol is responsible for reducing or mitigating the effects of cancer [3]. Carotenoids have a unique structure that produces numerous biological properties, including antioxidant, anti-inflammatory, and autophagy modulatory properties [4]. The phytosterols, including stigmasterol, brassicasterol, campesterol, ergosterol, lupeol, and sitosterol, can decrease lipid and cholesterol levels, preventing hypercholesterolemia, cardiovascular disease, and aging [5]. A general problem associated with pure bioactive compounds is the authentication and interpretation of bioactive molecules. The application of the plant genetic profile analysis is gaining inclusion in state and commercial analytical laboratories. The authentication of medicinal plants through DNA barcoding is a gold standard that has been successfully applied to detect adulteration in several high-value species. However, there are limitations within genetic profiling, and also in the accessibility of such technology in basic laboratory settings. In many cases the identification of key chemical biomarkers, active or purely analytical, is often the main approach used; this can be carried out through direct comparison with chemical reference compounds or by hyphenated methods, referring to public or commercial libraries of compounds. The choice of such identification or standardisation procedures is called into question by the fact that many phytochemicals are neither commercially available nor well characterised in databases. In addition, the phytochemistry of many herbal remedies can be quite complicated and is still mostly unknown. The idea of analytical profiling is then put out, with the goal of defining a reference fingerprint, chromatographic, spectroscopic, or chromato-spectrometric profile for a particular plant or extract, along with a record of its stability and acceptable deviations.

Natural product contamination, adulteration, and falsification all pose health problems due to their introduction of hazards or effects on bioactivity. The danger of these problems may be identified and reduced with the use of reliable foreign material detection techniques and efficient identification systems, both of which are essential for the purity and quality of pharmaceuticals. The identification and verification of the constituents of herbal medications are made possible by a number of taxonomic, chemical, proteomic, and genetic markers. These techniques include capillary electrophoresis, high-performance thin-layer chromatography, high-performance liquid chromatography, gas chromatography, morphological identification (macro- and microscopic identification), analysis of molecular markers, such as proteins or DNA, and analysis or profiling of secondary metabolites. All of these techniques can be combined with mass spectrometry [6].

Despite the availability of several natural products with potential pharmacological activities, the therapeutic efficacy in clinics is limited due to their low solubility, permeability, and bioavailability. Moreover, natural products are difficult to formulate because of their complex phytochemical components that arise as mixes with complex extraction techniques. Hence, the solubility, polymorphism, and particle size affecting biological activity should be considered to formulate bioactive compounds in appropriate, efficient dosage forms. To overcome these challenges, pharmaceutical manufacturers implemented different formulation strategies with various forms of excipients. Nanotechnology has been widely used to increase bioactive compounds’ physicochemical properties and therapeutic efficacy.

For most plant species with therapeutic activity, administering herbal preparations at the desired target is difficult in native form. Natural products such as tannins, terpenoids, and flavonoids that are soluble in water cannot cross biological membranes. For that reason, their absorption is less than that of other substances. Furthermore, their large molecular size reduces their bioavailability and efficiency [7]. Novel drug delivery technologies have been developed using plant-based bioactive compounds to reduce or eliminate limitations.

Nanotechnology is the engineering and manufacturing of materials at the atomic and molecular scale that provide choice for the selection of administrating drugs through different routes with enhanced therapeutic efficacy. Nanotechnology-driven herbal medicines offer potential and special qualities, such as turning less soluble and poorly absorbed ingredients, and incorporating unstable ingredients into promising pharmaceuticals. As a result, nanotechnology-based delivery methods offer a great opportunity to improve herbal activity and resolve problems using herbal therapy [8] (Figure 1).

Both hydrophobic and hydrophilic bioactive compounds have been successfully delivered using lipid-based nanoparticles, including liposomes, phytosomes, SLNs, and NLCs in clinical trials [9,10]. Doxil, the first approved by the FDA and nanotechnology-based drug delivery system, is a PEGylated liposome that contains doxorubicin (DOX) and is utilized in the management of carcinoma of the breast, ovary, and various other solid tumors [11]. Compared to free DOX, PEGylated liposomal DOX offers several advantages, such as a significantly lower risk of cardiotoxicity, a longer plasma retention time, and passive tumor targeting by utilizing the improved permeation and retention effect [12]. A significant turning point for lipid-based delivery technologies and anticancer nanomedicine was the clinical authorization of Doxil in 1995. Preclinical effectiveness of botanical compounds such as allicin, alpinumisoflavone, andrographolide, apigenin, baicalein, baicalin, decursinol, dicoumarol, emodin, genistein, gingerol, glycyrrhizin, hispidulin, nimbolide, pterostilbene, 6-shogaol, thymol, thymoquinone, ursolic acid, etc., have been explored in multiple animal models [13]. Numerous phytochemicals, particularly berberine, curcumin, green tea, and catechins, including epigallocatechin-3-gallate (EGCG), lycopene, quercetin, resveratrol, and sulforaphane, were demonstrated to be helpful in preclinical investigations. The phytochemicals berberine, curcumin, epigallocatechin gallate, quercetin, resveratrol, sulforaphane, and lycopene are now being tested in clinical studies for several malignancies [13].

Extracellular vesicles [EVs] are nano-sized, lipid bilayer-enclosed extracellular structures released from a variety of cell types into the surrounding environment. EVs are small vesicles with a membrane structure that are actively secreted by cells. They are mainly divided into three categories, which are exosomes, microvesicles, and apoptotic bodies according to their size, biological characteristics, and formation process. Recent studies have shown the potential of exosomes as delivery vesicles for bioactive compounds that may either be part of their cargo or lipid structure. Compared with conventional synthetic vehicles, EVs have several advantages, such as lower immunogenicity, longer circulation time, and better targeting capability. Moreover, extracellular vesicles such as biomimetic nanoparticles have recently emerged as a promising platform for bioactive molecules. Thus, they can serve as natural carriers for therapeutic agents and drugs, and have many advantages over conventional nanocarriers, including their low immunogenicity, good biocompatibility, natural blood–brain barrier penetration, and capacity for gene delivery. The combination of these innovative cell-derived nanotechnologies with the plant-derived actives might lead to the development of novel powerful therapeutic tools characterized by selectivity, low immunogenicity, and reduced side effects [14,15].

Previous reviews focused on use of herbal drug-loaded soft nanoparticles for treating skin disorders [16]; the present review provides collective information on the formulation of polymeric nanocarriers loaded with bioactive compounds such as paclitaxel, curcumin, rhodomyrtone, quercetin, kaempferol, resveratrol, epigallocatechin-3-gallate, silymarin, saponins, and oridonin, using advancement in technology for therapeutic improvement. Moreover, this review presents recent evidence on patented lipid-fortified products.

## 2. Lipid Based Nanoparticles

Liposomes are fabricated using phospholipids. The vesicle-based micelles are comprised of phospholipids containing one single layer, with the head group facing the outside and the hydrophobic tails forming the micelle core in a hydrophilic atmosphere, for example, blood, as opposed to liposomes, which are comprised of phospholipid bilayers comparable to cell membranes. The target drug’s physicochemical characteristics can guide the choice of liposomes used for certain delivery applications. For instance, the liposomal aqueous compartment fabricated by the hydrophilic head groups of the phospholipids includes one or more hydrophilic medicines. Thus, lipophilic medications are more appropriate for administration via micelles, in which the lipophilic tails of the phospholipids form the drug-containing compartment. However, liposomes are also used when a lipophilic drug disintegrates into a liposomal bilayer [17]. The hydrophobic core of solid lipid nanoparticles (SLNs) is surrounded by phospholipids, which is considered a substitute method for administering hydrophobic medications. In some circumstances, SLNs are less hazardous than polymeric nanoparticles and more stable than liposomes. Liquid lipids have been introduced in the solid structure to overcome several restrictions for old-generation SLNs, developing nanostructured lipid carriers [18]. The advantages of SLN over micelles are the ability to control drug release, improve stability of pharmaceuticals, and enhance drug-encapsulated content. In addition, SLNs retain excellent biocompatibility, use water-based technology to avoid using organic solvents, are easy to scale-up and sterilize, are more affordable, and are easier to validate and gain regulatory approval [19].

Phospholipid hydration has been used to prepare liposomal vesicles. Phospholipids are amphiphilic components and exhibit a hydrophilic head group and hydrophobic acyl chains in their structural makeup. This uniqueness makes it possible to hydrate phospholipids rapidly and efficiently. The hydrophilic groups of the phospholipid molecules combine to form an aquaphilic zone, and the lipophilic chains of the molecules confront one another to form a water-free region when they reassemble into a bi-layered structure in the aqueous phase. The hydrophilic groups in the vesicle are pointed to the aqueous phase, even though its hydrophobic tails are embedded within the phospholipid bilayer. With the aid of this structure, both hydrophobic and hydrophilic components can be trapped inside the aqueous core and lipid bilayer [20]. The phospholipids’ phase transition temperature is crucial in liposomal vesicular drug delivery. Phospholipids occur in a liquid crystalline phase above the transition temperature, where their hydrophobic tails are prepared to condense into liposomes. The hydrophobic tails of phospholipids become firmly packed when the temperature drops below the transition temperature, resulting in a gel state that prevents the formation of liposomes [21,22].

## 3. Lipids and Phospholipids Used in the Fabrication of Nanoparticles

Diacyl-chain phospholipids develop spherical or multilayered vesicles with a double layer of lipids once they self-assemble in aqueous solutions [20,23]. The fabrication of an amphiphilic structure is facilitated by the hydrophobic tail and hydrophilic head of the bilayer phospholipid membrane [24]. Both organic and synthetic phospholipids can fabricate lipid nanoparticles [25]. Particle size, stiffness, fluidity, stability, and electrical charge are only a few of the features of liposomes that are meaningfully affected by lipid content [26]. For instance, lipid nanoparticles made from naturally occurring unsaturated phosphatidylcholine, such as that found in eggs or soybeans, have extremely permeable and low-stable characteristics. However, liposomes made of saturated phospholipids, like dipalmitoyl phosphatidylcholine, produced hard, nearly impermeable bilayer structures [27]. Lipid hydrophilic groups contain zwitterionic or both positively and negatively charged molecules. Through electrostatic repelling, the hydrophilic group’s charge promotes stability [28]. Table 1 presents some of latest properties of commonly available and used lipids as drug-delivering carriers. Figure 2 illustrates the structures of various lipids used in the fabrication of SLNs. The lipids utilized to make lipidic nanoparticles fall into the following categories:

### 3.1. Natural Lipids

The normal cell membrane bilayer is primarily made up of glycerophospholipids. A glycerol unit is joined to two fatty acid molecules and a phosphate group to form phospholipids. The tiny, necessary choline chemical molecule can potentially be linked with the phosphate group [29]. Egg yolk and soy beans are two examples of primarily used natural phospholipid sources [30]. Phospholipids are classified into six classes based on their polar head groups: phosphatidic acid, phosphatidylethanolamine, phosphatidylcholine, phosphatidylserine, phosphatidylglycerol, and phosphatidylinositol [31]. Because the hydrocarbon chain of natural phospholipids is unsaturated, they are less stable than synthetic phospholipids when liposomes are produced [32]. Unsaturated fatty acids, such as margaric acid, are present in egg yolk lecithin, a fatty acid in natural phospholipids [33].

### 3.2. Synthetic Lipids

Natural phospholipids’ polar and non-polar portions are chemically modified to develop synthetic phospholipids. The alteration allows for an infinite number of phospholipids that can be characterized and categorized [34]. PEGylated phospholipids and cationic phospholipids 1,2-diacyl-P-O-ethylphosphatidylcholine are common examples of synthetic lipids that have been further investigated for pharmacologic efficacy by converting them into a prodrug form [34]. Moreover, an investigation indicated that bilayer-forming synthetic lipids such as dioctadecyldimethyylammonium and dihexadecylphosphate salt-fabricated vesicles that interact using prokaryotic or eukaryotic cells, which are negatively charged, yield adsorption isotherms of strong affinities towards the cell membrane, initiating cell adhesion as well as flocculation, transforming the cell surface to positive from negative with significant effects on viability [35]. In another study, Lin and co-workers fabricated phosphatidylcholine vesicles incorporating a novel lipid/poly-phosphocholine conjugate, which stabilized the liposomes against aggregation by acting as efficient lubrication via hydration at a physiological salt concentration and pressure, suggesting this modification results in stable super lubrication vectors in potential biomedical applications [36].

### 3.3. Steroids

Steroids are hydrophobic lipids with a four-ring structure. The different functional groups connected to those rings provide steroids with diversity. Due to being integrated into the lipid bilayer of nanoparticles, cholesterol is the main steroid typically employed in nanoparticle synthesis at a fraction below thirty percent of the total lipid content to enhance liposome firmness and robustness. Researchers compared the effects of cholesterol and β-sitosterol on the membrane features and discovered these two steroids decreased the fluidity of the membrane, increased the absolute zeta potential, significantly changed the particle size, and decreased the temperature and enthalpy of the dipalmitoyl phosphatidylcholine phase transition [37].

**Table 1 pharmaceuticals-17-00329-t001:** Commonly available and used lipids as drug-delivering carriers.

Lipid Carrier	Drug Delivery	Bioactive Compounds	Preparation Method/Composition	Critical Attributes and Outcomes	Reference
Dipalmitoyl phosphatidylcholine	NLC-skin	Quercetin	Melt emulsification method followed by ultrasonication	Higher drug loading capacity makes them more suitable for topical skin administration and ensures longer colloidal stability which also shows a positive impact on skin permeability.	[38]
Soy-l-α-phosphatidylcholine	Injectable liposomes	Paclitaxel	Cell disruptor-type sonicator	Improved bioavailability, solubility, biodistribution, and intracellular uptake of Paclitaxel.	[39]
1,2-distearoyl-sn-glycero-3-phosphoethanolamine	Liposomes	Naringin	Thin-film hydration	Improved solubility as well as dissolution, enhanced permeability and retention effect in Rheumatoid Arthritis treatment	[40]
Glycerol monostearate and medium-chain triglycerides	NLC	Curcumin	Emulsification	The encapsulation of curcumin in an NLC was increased, resulting in an increase in its antimalarial activity as well as increased colloidal stability.	[41]
Egg phosphatidylcholine	Intranasal Liposomes	Quercetin	Thin film hydration	Intranasal liposomes showed better solubility and dissolution at a lower dose than the other treatments. The best cognitive and anxiolytic effects for intranasal QU liposomes may be attributed to the alteration of various neurotransmitters.	[42]
Dipalmitoylphosphatidylcholine and Dipalmitoyl-sn-glycero-3-phosphoglycerol	Liposomes	curcumin	Conventional thin film hydration technique	Curcumin-loaded liposomes exhibited sustained release and substantial antibacterial activity against Gram-positive bacteria.	[42]

## 4. Natural Bioactive Compounds in Lipid-Based Drug Delivery Systems

### 4.1. Paclitaxel

A naturally derived bioactive compound, diterpenoid pseudoalkaloid, recognized as paclitaxel, was discovered in the bark of the Pacific Yew (*Taxus brevifolia*) [43] and is frequently employed as a form of chemotherapy in treating various cancers, including progressive lung, breast, and ovarian cancers [44]. The stabilization of micro-tubule blockage of the cancer cell cycle during the G2/M phase and eventually triggering apoptosis is how paclitaxel works [45]. The insufficient solubility of paclitaxel in aqueous media (0.4 g/mL) is the main barrier to administration. The commercially available preparation (Taxol^®^) overcame this flaw by solubilizing paclitaxel in chromophore EL/ethanol (1/1, *v*/*v*). However, chromophore EL, has been linked to several serious side effects, notably heightened sensitivity, cardiac toxicity, and neuropathy in the extremities. Moreover, the cellular uptake investigation indicated that coumarin-6-loaded nanoparticles and nanoparticle surfaces modified with polydopamine were densely dispersed around the nuclei (as indicated by DAPI staining), implying that the fluorescent nanoparticles had been internalized into A875 cells. Furthermore, the fluorescence of coumarin-6-loaded nanoparticle surfaces modified with polydopaminein in the cytoplasm was much brighter than that of cumarine-6-loaded nanoparticles, indicating facilitation of the internalization of cationic nanoparticles by negatively charged membranes (Figure 3) [46].

For the targeted delivery of paclitaxel, Zhao and colleagues formulated and assessed an effective carrier made of a polymer core of nanoparticles with a lipid shell modified by folic acid. This formulation included a PLGA core, a PEGylated octadecyl-quaternized chitosan lipid shell, and cholesterol. The biocompatibility investigation of paclitaxel-loaded folic acid nanoparticles indicated improvement compared with paclitaxel formulation used in marketplace formulations (Taxol^®^). Compared with Taxol^®^, intravenous delivery of paclitaxel-encapsulated folic acid nanoparticles reduced tumors and increased the lifespan of animals in a mouse model [47]. A biodistribution investigation also revealed that paclitaxel-enclosed folic acid nanoparticles had an improved paclitaxel concentration in tumors compared with other paclitaxel products, suggesting effective nanosized therapeutic preparation of folic acid nanoparticles to treat malignancies [48].

To formulate an intravenous dosage form without chromophore EL for the safety and clinical efficacy improvement of paclitaxel, Danhier and colleagues developed paclitaxel-loaded PEGylated PLGA nanoparticles using a nano-precipitation technique [49]. In vitro anticancer activity towards human cervical carcinoma cells altered the release pattern and lowered IC_50_ (5.5 g/mL), as opposed to commercially available products (15.5 g/mL), with concentration-influenced cellular absorption and equivalent apoptosis indicated by the fabricated nanoparticles.

Dattani et al. fabricated lipid-based formulations of paclitaxel using solvent evaporation, thin-film hydration, and microfluidic mixing. Further, they compared the in vitro and in vivo characteristics to provide further insights into their therapeutic potential for clinical development. The investigation indicated that nanosized vesicles incorporated paclitaxel with the slowest release rate and highest stability in biological media for SLNs, compared to other formulations. Furthermore, the SLNs incorporated statistically delayed tumor growth, compared to mice treated with liposomes and micelles, indicating the potential efficacy of delivering and managing cancer using paclitaxel [50]. Generally, SLNs combine the advantages and avoid the drawbacks of several colloidal carriers of its class, such as controlled drug release, improving the stability of pharmaceuticals, and high and enhanced drug content. In addition, SLNs retain excellent biocompatibility, use water-based technology to avoid the use of organic solvents, are easy to scale-up and sterilize, more affordable, and easier to validate and gain regulatory approval. Moreover, apoptosis of cancer cells was increased (45%) by Duan et al. who developed paclitaxel and etoposide-loaded lipid–polymer hybrid nanoparticles (PE-LPN) and compared them with all additional groups that had superior caspase-3 and -8 activities. In cancerous tissues, the nanoparticles demonstrated an ordinary endocytosis-mediated uptake by the cells. Notably, PE-LPN demonstrated an impressive tumor regression impact and two-fold higher effectiveness over native forms [51].

Godara and colleagues developed a nanoprecipitation process to develop paclitaxel-loaded polymeric nanoparticles employing poly lactic-co-glycolic acid as the polymer and stearyl amine and soy lecithin as the lipids with stable structures. Stabilizers were fabricated using surfactants, including polyvinyl alcohol, tocopheryl polyethylene glycol succinate, Pluronic 68, and human serum albumin (HSA). A viable nano-system for the regulated distribution of paclitaxel was suggested by the synthesized nanoparticles mixed lipid coating of PLGA with HSA [52]. In an additional investigation, Liu and colleagues fabricated nanoparticles with a lipidic monolayer and a biodegradable polymeric core to regulate the release of paclitaxel [53]. Particular focus was placed on the impacts of the surfactant used and the optimization of the emulsifying agent employed in the single emulsion solvent evaporation technique. When forming nanoparticles of biodegradable polymers like PLGA, it was discovered that phospholipids with short chains, such as 1,2-dilauroylphosphatidylocholine (DLPC), had several benefits over the traditional emulsifying agent poly(vinyl alcohol) (PVA). Following treatment with coumarin-6-loaded PLGA nanoparticles of DLPC shell against MCF-7 cells at a concentration of 0.25 mg/mL, the DLPC nanoparticle cell absorption was larger than that of the PVA shells. Additionally, the IC_50_ value showed that DLPC shell PLGA core nanoparticles of paclitaxel were seven-fold more potent after prolonged treatment for three days, compared with the marketed product Taxol^®^ [53].

### 4.2. Curcumin

A naturally occurring compound, curcumin, is obtained from the rhizomes of turmeric (*Curcuma longa*), which has been employed medicinally for thousands of years [54]. Curcumin has been reported to manage metabolic syndrome, arthritis, anxiety, hyperlipidemia, and oxidative and inflammatory diseases [55]. Additionally, it aids in controlling inflammation and discomfort in the muscles brought on by physical activity, improving regeneration and effectiveness among individuals. Furthermore, investigations have also demonstrated several pharmacologic effects, such as anti-diabetic [56], anti-inflammatory [57], and antioxidant [58] properties. Numerous positive effects, including anti-hepatotoxic [59], anti-proliferative [59], anti-carcinogenic [60], anti-microbial [61], neuroprotective [62], and anti-aging [62] capabilities are currently documented. Although it appears to have a wide range of medicinal properties, it has restricted applicability because of its inadequate water-soluble nature, bioavailability, quick biotransformation, inadequate permeation or absorption, instability in alkaline environments as well as light, and sensitivity to metallic ions [63]. The bio-accessibility, absorption, distribution, and elimination of curcumin are constrained in the context of oral active-loaded methods of administration [64]. Investigators have developed lipid-based administration methods to increase the oral bioavailability of curcumin.

Nanocurcumin formulated using phospholipids derived from soy lecithin and cholesterol had an average size of 136.4 ± 3.44 nm. The particle size distribution was controlled using PEO-PPO-PEO triblock-co-polymer poloxamer 407 as a nonionic surfactant and polyethylene glycol 400. The encapsulation efficiency of curcumin extract was 90.74 ± 5.35% in nanocurcumin. The in vitro release content of curcumin from nanocurcumin was higher than that of curcumin extract, indicating that PEG and poloxamer-coated nanocurcumin can maintain its dispersion stability [65].

Quirós-Fallas and colleagues fabricated a composite fortified with lipid nanoparticles incorporating curcumin and showed effective antioxidant properties and immune-stimulating effects with biocompatibility impacts on malignant cell lines [66]. In another study, curcumin nanoparticles were formulated, demonstrating superior anti-bacterial properties against potential pathogenic microorganisms and the apoptosis of cells [67]. By triggering apoptosis in cells due to caspase activation, inhibiting NF activation, and upregulating TNF-R for SLNs when in comparison with unbound curcumin in human carcinoma cell lines, Vandita et al. demonstrated a 32- to 155-fold rise in bioavailability for curcumin when used alongside the SLNs (Figure 4) [68]. Nair and colleagues fabricated curcumin-loaded chitosan nanoparticles by ionic gelation to enhance transdermal penetration and discovered that a large chitosan to tripolyphosphate ratio results in bigger particles. However, tripolyphosphate facilitated chitosan, indicating stronger intramolecular associations from smaller particles [69].

Furthermore, curcumin-loaded SLNs of fatty acids were formulated by Chirio et al., employing a coacervation process, revealing improved encapsulating effectiveness ranging from 28 to 81% [70]. To permeate curcumin topically, Caon and colleagues developed SLNs, nanoemulsions, and nanoparticles of polymers. The dermal penetration of curcumin using Franz diffusion cells through pig ear skin indicated that SLNs, nanoemulsion, and nanoparticles made of polymeric material considerably improved bioactive compound accumulation in the outer layer of skin by 2.49-fold and 3.32-fold, respectively [71]. In a recent study, vesicle-incorporated curcuminoids fortified within a biodegradable composite demonstrated improved stability and therapeutic efficacy, indicating that vesicle stability can be significantly enhanced through this process [72].

### 4.3. Rhodomyrtone

A plant-based antibiotic called rhodomyrtone is found in the whole plant *Rhodomyrtus tomentosa* (Myrtaceae) [73], demonstrating substantial anti-bacterial effectiveness towards numerous drug-resistant strains of Gram-positive bacteria such as *Enterococcus faecalis*, *Propionibacterium acnes*, *Staphylococcus aureus*, *Streptococcus pneumoniae*, and *Streptococcus pyogenes* [74]. Liposomal-encapsulating rhodomyrtone composed of phosphatidylcholine and cholesterol was developed by Chorachoo and colleagues, who demonstrated minimal inhibitory concentration and minimal bactericidal concentration values. The MIC and MBC were reported from 1 to 4 and 4 to 64 g/mL against bacteria isolates and reference species, respectively, while those of rhodomyrtone were between 0.25 and 1 and 0.5 and 2 g/mL, respectively. These results proved that the effectiveness of liposome formulations was greater than that of taking a single drug [75]. Recently, in another investigation, rhodomyrtone-rich *R. tomentosa* extract-incorporated lipid vesicle-stabilized hydrogel indicated an improvement in the stability as well as antimicrobial efficacy, demonstrating that hydrogel transformation could be a suitable method for the stabilization of such vesicles [76].

### 4.4. Quercetin

Quercetin (3,3′,4′,5,7-penthydroxyflavone) is a food supplement (polyphenolic substance) that is found in a variety of foods, such as apples, buckwheat, onions, and citrus fruits [77]. Investigations showed quercetin exhibits medicinal properties and health effects, including antioxidant, anti-inflammatory, antiplatelet, anti-apoptotic, nephron, gastro-, angio-, cardio-, and chondroprotective effects, so quercetin has become prevalent in medicine [78]. Polymeric nanocarriers could serve as a useful technique for encapsulating quercetin due to their ability to improve bioavailability, pharmacologic potency, and in vivo resilience.

Shawky and colleagues fabricated an intravesical system administering quercetin to treat bladder carcinoma for enhanced therapeutic effectiveness. Quercetin-loaded SLNs that were either uncoated or coated with chitosan showed concentrations depending on the toxic effect, with a quercetin IC_50_ of 1.6 to 8.9 μg/mL. Compared with uncoated SLNs, coated SLNs showed improved penetration capabilities, while coated SLNs dispersed in gel demonstrated maximum permeation (Figure 5) [79]. Yin and colleagues developed cholate-modified polymer–lipid hybrid nanoparticles to enhance the antileukemic effect of quercetin. The developed quercetin nanoparticles resulted in a substantial boost in bioavailability, approximately 3.7-fold, compared with the product of suspensions with excellent cell absorption and internalizing capacity compared with cholate-free nanoparticles [80]. In another investigation, quercetin-loaded mesoporous silica nanoparticles were developed and optimized by Elmowafy and colleagues to enhance quercetin’s dissolution rate and bioavailability. The optimized nanoparticles improved the saturation solubility and dissolution rate while displaying satisfactory physiochemical characteristics [81]. Quercetin-loaded PLGA nanoparticles were prepared using hyaluronic acid (HA) as it has mucoadhesive properties and may increase the retention of the nanoparticles in the cornea [82]. HA significantly enhanced the colloidal stability due to electrostatic and steric stabilization. The quercetin in the PLGA matrix decomposed via a second-order reaction.

Resistance to many drugs is a significant barrier to the medicinal application of chemotherapeutic medications; their main drawbacks with free medications are poor water soluble, physiochemical, and pharmacologic behavior. Nano-formulations employing natural biopolymers can overcome the constraints of free medicines. Quercetin was used as an anticancer therapeutic, encapsulated within lipid nanocarriers and delivered through the P-glycoprotein receptor by Kumar et al. to treat breast cancer. Compared with unloaded quercetin, the quercetin-loaded SLNs demonstrated augmented cytotoxicity in drug-resistant human breast carcinoma cell lines [83].

Patil and Mahajan developed quercetin-loaded nanostructured lipid carriers to target the brain for nasal administration. Wistar rats were employed to investigate the passage of drugs through the nose to the brain. Compared with quercetin, prolonged drug administration and considerable brain targeting were accomplished using nanostructured lipid carriers, suggesting that delivering bioactive compounds through nose-to-brain could potentially be accomplished using nanostructured lipid carriers [84]. Moreover, employing glyceryl behenate as a lipidic component and poloxamer 188 as a surfactant molecule, Hariyadi and colleagues manufactured a solid lipid microparticle system encapsulating quercetin. Reduced poloxamer concentrations resulted in burst releases of quercetin, while higher surfactant concentrations resulted in a prolonged and controlled release profile [85]. Furthermore, quercetin-incorporated liposomes fabricated by Tefas and colleagues demonstrated that DPPC (Di palmitoyl phosphatidyl choline) level and lipid ratio were the primary determinants of particle size. In contrast, quercetin content primarily influenced entrapment efficiency [86].

### 4.5. Kaempferol

Kaempferol is a flavonoid, which are regarded as the largest group of secondary plant metabolites [87]. The therapeutic effects of kaempferol include anti-bacterial, anti-inflammatory, antioxidant, anticancer, cardioprotective, neuroprotective, anti-diabetic actions, and anticancer effects [88]. Kaempferol-loaded nanoparticles were developed using the quasi-emulsion solvent diffusion approach, and their oxidative and liver-protective activities were examined using hepatocellular cancer models by Kazmi and colleagues. Kaempferol coated with Kollicoat and hydroxypropyl methylcellulose acetate succinate nanoparticles demonstrated a significant reduction in interleukin-1β, IL-6, and tumor necrosis factor-α and nuclear factor kappa-light-chain-enhancer of activated B cell protein production. However, the heme oxygenase-1 and nuclear factor erythroid 2-related factor 2 had both their mRNA and protein expression levels increased.

Moreover, kaempferol nanoparticles were shown to be superior to pure kaempferol. Additionally, increased antioxidant enzymes and proteins and substantially reduced blood levels of various liver biomarkers were discovered [89]. In another investigation, *Kaempferia galanga* rhizome oil with major constituents of cinnamate fabricated as a microemulsion using tween 80 and surfactant fortified within hydrogel revealed mild in vitro sun-protective and anti-inflammatory effects towards lipopolysaccharide-stimulated macrophages.

He and colleagues fabricated kaempferol nanosuspensions (KAE-NS) using D-α-Tocopherol polyethylene glycol succinate (TPGS) as a stabilizer to improve the anticancer efficacy of kaempferol. The results indicated significantly improved entrapment efficiency and solubility of kaempferol, compared with kaempferol cryopreserve using glucose. In addition, TPGS-KAE-NS exhibited favorable stability and biological compatibility, contributing to a sustained release profile. A greater cytotoxicity, cell migration, and intracellular generations of ROS were demonstrated by TPGS-KAE-NS in vitro cell tests compared with those of pure kaempferol [90]. To tackle the bioavailability issue, Ilk and colleagues fabricated lecithin-chitosan nanoparticles (LCNPs) of kaempferol to evaluate antifungal efficacy against the phytopathogenic fungus *Fusarium oxysporium*, compared with pure kaempferol. Kaempferol was successfully encapsulated within LCNPs with remarkable physicochemical stability [91].

### 4.6. Resveratrol

Resveratrol (3,5,4′-trihydroxy-trans-stilbene) is in the stilbenoids family of polyphenols found in a variety of plant species, including grapes, peanuts, and legumes [92]. Resveratrol chemically exists in two isomers, cis-(Z) and trans-(E) isomers, with the trans-isomer being regarded as more stable and physiologically effective compared with the cis-isomer [93]. Nevertheless, when exposed to ultraviolet rays, the trans-isomer changes to the cis-isomer form [94]. Numerous biological effects of resveratrol, including cardiovascular protection, cancer inhibition, platelet de-aggregation, and anti-inflammatory activity with vasorelaxant characteristics, were reported [95]. Additionally, it could enhance the antiviral effects of didanosine, zalcitabine, and zidovudine [96]. However, resveratrol indicates several major biological drawbacks of the medication reported regarding its water solubility [97].

SLNs were developed by Sun et al. as workable means of delivering drugs to get around a few of these problems and broaden their applicability. The findings demonstrated the unanticipated involvement of resveratrol SLNs in food deprivation-induced mitochondrial oxidant generation by mitochondrial quality control. Resveratrol SLNs probably engaged a mitochondrion-mediated mechanism that enhanced endurance (Figure 6) [98]. Moreover, resveratrol liposomes were developed by Bojana and colleagues, employing various methods, and were assessed for several parameters related to the process. The thin-film approach and pro-liposomes were discovered to have maximum entrapment performance. Additionally, the particle size decreased using extrusion and sonication techniques; nevertheless, compared to the sonication method, it entrapped greater resveratrol [99].

Employing hydroxypropyl methylcellulose as a stabilizer using heat-controlled anti-solvent precipitating techniques, Kim et al. developed resveratrol nanoparticles. It has been discovered that the average particle size dropped as the precipitation temperature changed due to higher nucleation speed and a lower growth rate [100].

### 4.7. Epigallocatechin-3-gallate

Epigallocatechin-3-gallate, in particular, is a phenolic substance derived from catechin, which is extremely common in tea, cocoa, and berries [101]. As one of the most potent bioactive compounds among tea’s polyphenolic compounds, EGCG is potentially linked with the anticancer properties of green tea [102].

Zhang et al. developed EGCG-encapsulated nanostructured lipid carriers (NLC) coated with chitosan. The investigation significantly indicated improvement in the stability of EGCG via nanoencapsulation, compared to unencapsulated. Moreover, encapsulated and chitosan-coated EGCG prominently reduced macrophage cholesteryl ester levels to 10 μM [103]. SLNs functionalized with folic acid were tested against breast cancer cell lines, and human mammary epithelial cells indicated a considerable cytotoxic effect at low tested dose; however, no effect on the survival of MCF10A cells was observed (Figure 7) [104].

Hajipoura and co-workers fabricated arginyl-glycyl-aspartic acid (RGD)-incorporated EGCG NLC and tested apoptotic induction. The results demonstrated that EGCG- NLC-RGD had the highest level of apoptotic effect with more rapid cell cycle inhibition compared with EGCG by the attachment of αvβ3 integrin [105]. Moreover, EGCG-grafted-chitosan nanoparticles were fabricated to maximize the therapeutic efficacy to reduce the minimum inhibitory concentration against *Pseudomonas fluorescens*. Additionally, nanoparticles increased the antioxidant activities and reactive oxygen species in two different bacterial strains [106]. Additionally, to enhance EGCG effectiveness towards atherosclerosis, EGCG incorporated self-organized chitosan and aspartic acid nanoparticles, indicating equivalent lipid deposition efficacy to simvastatin [107]. In another investigation, Liang and colleagues developed chitosan/-Lactoglobulin nanoparticles loaded with EGCG to deliver and prolong the release of EGCG in the digestive canal [108].

### 4.8. Silymarin

The dehydrated seeds and fruits of the milk thistle shrub (*S. marianum*) are known as the origin of silymarin. Since ancient times, the milk thistle has been traditionally utilized medically in Europe [109]. The silymarin found in milk thistle is a complex blend of plant-derived substances, the majority of which have been classified as flavonolignans, flavonoids (taxifolin, quercetin), and polyphenolic compounds [110]. It was utilized as an alternative medicine to preserve the liver and treat several liver conditions, including cirrhosis and hepatitis. Therapeutically, it possesses anti-inflammatory, antifibrotic, anti-lipid peroxidative, and antioxidant effects. In addition, it has received extensive research as a chemopreventive drug for many malignancies [111].

To improve silymarin bioavailability, Liang and co-workers developed silymarin-loaded lipid–polymer hybrid nanoparticles of chitosan (CS-LPNs) and assessed their impact on lipid levels. According to the findings, silymarin from CS-LPNs was 14.38 times more bioavailable than that obtained through suspensions. Additionally, compared to chitosan-free nanoparticles, CS-LPNs significantly improved the liver’s functioning by reducing the liver’s lipid and blood lipid levels (Figure 8) [112].

To enhance the in vitro antitumor applicability, Snima and colleagues developed PLGA-based nanoparticles of silymarin using a single-step emulsion process. The PLGA-nanoparticles that encapsulated over 60% of the silymarin compound induced time- and dose-specific cytotoxic responses towards prostatic tumor cells [113]. Hirlekar and co-workers fabricated NLCs of silymarin by glyceryl monostearate, capmul MCM C8 EP, and gelucire 50/13 as the appropriate solid lipid, liquid lipid, and surfactant using improved hot melt emulsification and ultrasonication. However, the ex vivo permeability with and without a lymphatic inhibitor demonstrates lymphatic absorption of silymarin-loaded NLCs [114].

To enhance anti-inflammatory efficacy towards sepsis and septic shock, Azadpour and colleagues developed PLGA nanoparticles of silymarin, which demonstrated improved anti-inflammatory cytokines, M2-associated markers, and proteins with significantly reduced pro-inflammatory cytokines [115]. In another investigation, a pH-responsive alginate-PLGA nano hydrogel-fortified silymarin nanoparticle developed by Elsherbiny and colleagues indicated controlled release, and improved oral bioavailability and dissolution [116].

### 4.9. Saponins

Saponins are naturally available bioactive compounds found in large quantities throughout all the cells of legume plants. Saponins are a complex and diverse collection of chemicals that can develop stable and soap-like foams in aqueous solution. Saponin glycosides have demonstrated a significant cytotoxic effect on several cell lines [117]. To manage and prolong distribution to prostatic carcinoma cell lines, Sanoj and colleagues developed saponin-fortified chitosan nanoparticles that can enhance the absorption of the compound in the cellular internalization experiments (L929 and PC3 cell lines) using a selective cytotoxic effect targeting malignant cells [118]. To enhance medication delivery in Leishmania-infected macrophages, Saharan and colleagues fabricated saponin β-aescin-loaded PLGA nanoparticles with a negative zeta potential and a particle size of less than 300 nm, which were shown to be significantly engulfed by the macrophages and transported to the lysosomes [119].

### 4.10. Oridonin

*Isodon rubescens* (Hemsl.) H. Hara, an ancestral Chinese herb, contains oridonin as one of the natural terpenoids [120]. Several reports indicate medicinal properties such as effects against malignant cells [121], anti-inflammatory effects [122], hepatoprotective function [123], cardioprotective action [124], lung protective effects [125], protective effects on neurons [126], and others. However, its usage is restricted due to oridonin’s poor water solubility and constrained therapeutic index. To extend PLA nanoparticle retention in the plasma and to enhance drug delivery, Xing and colleagues developed oridonin-loaded PLA nanoparticles, which displayed an excellent improvement in therapeutic efficacy for a prolonged duration. They also revealed considerable oridonin deposition in the spleen, liver, and lung; however, the concentration in the kidneys and the heart was significantly reduced [127]. Utilizing stearic acid, soybean lecithin, and pluronic F68, Zhang and colleagues prepared SLNs of oridonin [128]. Oridonin-loaded SLNs evidently improved the bioactive compound content in the liver, lung, and spleen, whilst decreasing its circulation in the heart and kidney, according to in vivo tests.

## 5. Safety Evaluation of Lipid-Based Nanoparticles Encapsulating Bioactive Compounds

Although some investigation indicates that lipid-based nanocarriers are used in the pharmaceutical and food industries, safety should not be taken for granted from the consumers’ point of view. Because their properties are distinct from bulk substances, it is hypothesized that nanocarriers with characteristics like nano-size, large surface-to-volume ratios, and effective infusion through the gastrointestinal tract into the bloodstream could trigger cytotoxicity [129]. Upon reaching the bloodstream, interacting with tissues, and producing adverse reactions, nanoparticles may cause oxidative harm, inflammatory conditions, and animal DNA damage [130]. Growing data have shown that nanoparticles could induce autophagy and may additionally result in nanotoxicity [131]. Concerns about alarming bio-persistent solid nanoparticles entering the dermis and potentially damaging living epidermal cells or accumulating in secondary organs after biological distribution have also been raised [132]. It is fortunate that publications on the assessment of nanocarriers’ safety are growing too. An increasing corpus of research indicates that specific bioactive material-loaded nanocarriers are effective and secure for preparations for application in foodstuff or pharmaceuticals. Thermosensitive irinotecan-encapsulated SLNs were effortlessly administered to the anus, and gel formed quickly and firmly following rectal delivery, causing no harm to the rat rectum and not causing a reduction in weight in tumor-carrying mice [133]. These characteristics included gel attributes, pharmacokinetics, morphology, effectiveness against cancer, and immune-histopathology. A more recent study demonstrated that several nanocarriers, including the bio-nano-capsule-liposome nanocarrier [134], poly(benzyl-L-aspartate)-co-[N-(3-aminopropyl) imidazole-L-aspartamide]-poly(ethylene glycol) [135], polysaccharide-based carriers [136], and poly(aspartic acid-graft Epsilon-caprolactone polymerized with poly(ethylene glycol) [137], were shown to be safe in vivo when being used to treat intestinal inflammation and malignancies. Although a wide range of nanocarriers are being developed, several of them are currently going through sophisticated clinical testing. Few studies, meanwhile, have fully translated research into therapeutic practice. This sluggish adoption could be partially explained by the necessity for a thorough proof of safety for these novel nanotechnologies. While the appropriateness (or the safe and effective dosage) of most food-related substances has been developed, more research remains to be carried out on the risk associated with pharmaceuticals and food produced by nanotechnology. Rats that consumed biodegradable polymers chitosan-sodium alginate-oleic acid nanocarriers infused using lutein at a dosage of 10 mg/kg of body weight showed no signs of fatality and no morphological or pathological alterations [129].

### 5.1. Preclinical and Clinical Aspects of Lipid-Based Nanoparticles Encapsulating Bioactive Compounds

Investigation is necessary to determine the proper dosage of a bioactive compound ingredient in an effective pharmaceutical product. It frequently takes the shape of preclinical and clinical investigations using human and animal participants [138]. The minimum dose at which a bioactive compound can influence the biomarkers of an illness needs to be tested in these experiments at various levels. Investigators must watch for potential side effects whenever participants receive dosages exceeding the crucial level. Clinical investigation is utilized to establish the ideal dosage of a bioactive compound to evaluate the effectiveness and safety of that biologically active substance. Additionally, there are several approaches to preclinical and clinical testing. Generally, preclinical and clinical trials are crucial for determining the correct amount of a bioactive compound.

Several distinct biologically active compounds found in herbs are derived from medicinal herbs. Environmental, harvesting, drying, preservation, and manufacturing processes can all affect the levels of chemicals. The three most important characteristics of synthetic and natural medications in treatment are efficacy, safety, and quality. Mostly, standardized extracts did not indicate any hazardous contaminants and possessed a particular quantity of bioactive molecules. The standardized extracts’ therapeutic, prophylactic, and preventative effects are linked to their complex chemical makeup. Toxicology testing, dosage response correlations, and clinical trial studies for natural products’ various active constituents are perplexing and hard to understand. The ineffectiveness of preclinical investigations on herbal products, their standardization, and the truth or falsity of or difficulties with their clinical investigations need to be assessed. In addition to being used in preclinical investigations and clinical frameworks, lipid-based nanotechnologies are being extensively exploited for drug delivery systems.

### 5.2. Patented Lipid-Based Nanoparticles Encapsulating Bioactive Compounds

Lipid-based nanoparticles encapsulating bioactive compounds have gained significant attention in various fields, particularly pharmaceuticals and drug delivery. These nanoparticles serve as carriers for delivering bioactive compounds, such as bioactive compounds, genes, or other therapeutic agents, to specific target sites in the body. The encapsulation of these compounds within lipid-based nanoparticles offers several advantages, including enhanced bioavailability, stability, and controlled release. Several companies and researchers file patents on developing and using lipid-based nanoparticles to encapsulate bioactive compounds. These patents often cover specific formulations, methods of preparations, and applications. Table 2 lists the major categories of lipid-based nanoparticles that are the focus of patent applications.

## 6. Conclusions, Future Prospects, and Challenges

There are several ways to distribute phytochemicals to the desired target. The stability of phytochemicals during large-scale manufacturing is of significant concern, and regulatory considerations for chemotherapy when used alongside synthetic drugs also need to be revised. Various delivery systems using nanomedicine or nanocarrier-based techniques are yielding encouraging results. However, nanocarriers may exhibit some level of harm in humans based on commercial or therapeutic possibilities, even if this is unproven. Although the smaller particles may overcome cell walls, there is little possibility that a substance will deposit in the body and be removed later. Despite being favored, the oral route has serious limits when administering phytochemicals because of solubility, acidity, enzymatic degradation, and the first-pass effect, all of which must be carefully considered. The phytochemical-based nano-formulations are typically administered intravenously or topically in preclinical models. Nanoparticles have been considered to have superior pharmacological and physicochemical properties because of things like their improved stability, capacity to stop drug leaks, simplicity in surface functionalization, and ability to be transformed into intelligent delivery systems. We also shed light on clinical translation challenges and how to solve them. Theranostics, auto-immune illnesses, the disruption of biological barriers, and other topics have received little attention, even though substantial research has been carried out in some areas like cancer and microbial biofilms. Researchers must concentrate their efforts in these areas to explore uncharted territory and use the developments in nanoparticle technology to develop the present therapies in the disciplines.

## Figures and Tables

**Figure 1 pharmaceuticals-17-00329-f001:**
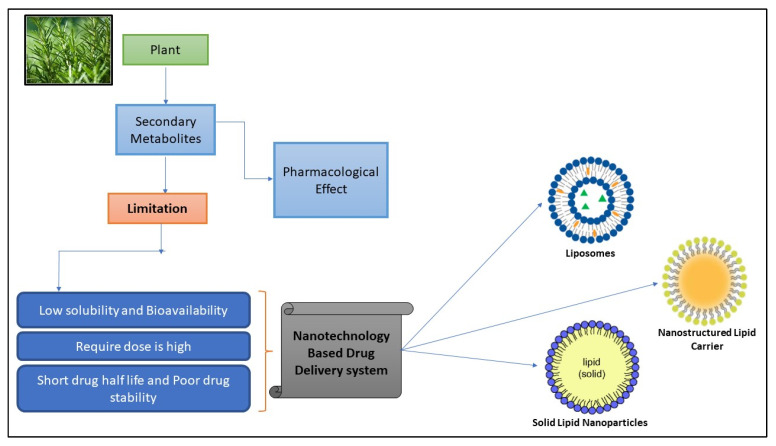
Diagram illustrating the flaws of bioactive compounds with their problem and solution.

**Figure 2 pharmaceuticals-17-00329-f002:**
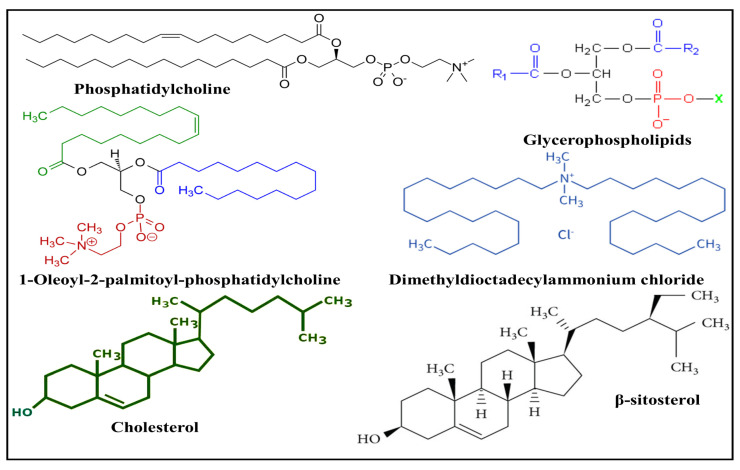
Illustration of chemical structure of different types of lipids used in fabrication of SLNs.

**Figure 3 pharmaceuticals-17-00329-f003:**
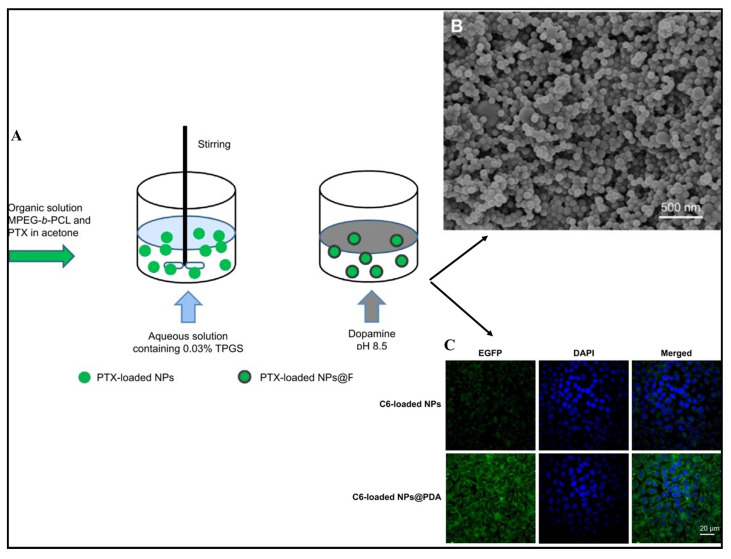
Graphical representation indicating fabrication technique to develop nanoparticles (**A**), scanning electron microscopy of bioactive compound-fortified nanoparticles (**B**), and confocal microscopy of bioactive compound-incorporated nanoparticles after 2 h of incubation in A875 cells. The fluorescence images were taken in the presence of emitting light from a source where EGFP is an enhanced green fluorescent protein, and DAPI is 4,6-diamidino 2-phenylindole dihydrochloride (**C**). Reproduced with permission from [46].

**Figure 4 pharmaceuticals-17-00329-f004:**
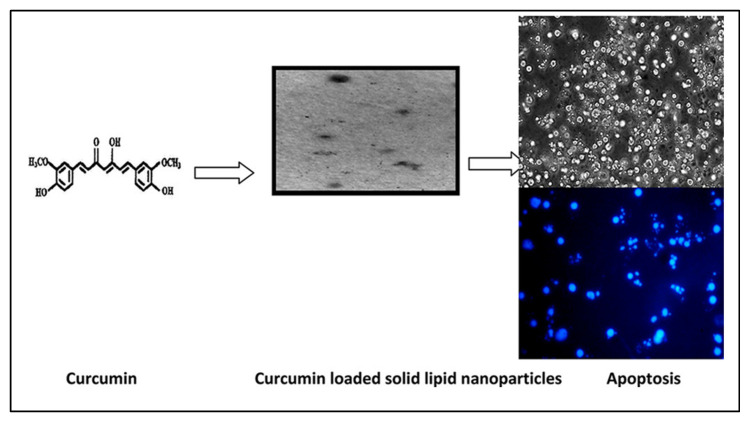
Schematic illustration indicating cancer cell apoptosis in the presence of solid lipid nanoparticles fortified with curcumin. Reprinted with permission from [68] under the Creative Commons attribution (CC BY 4.0) license.

**Figure 5 pharmaceuticals-17-00329-f005:**
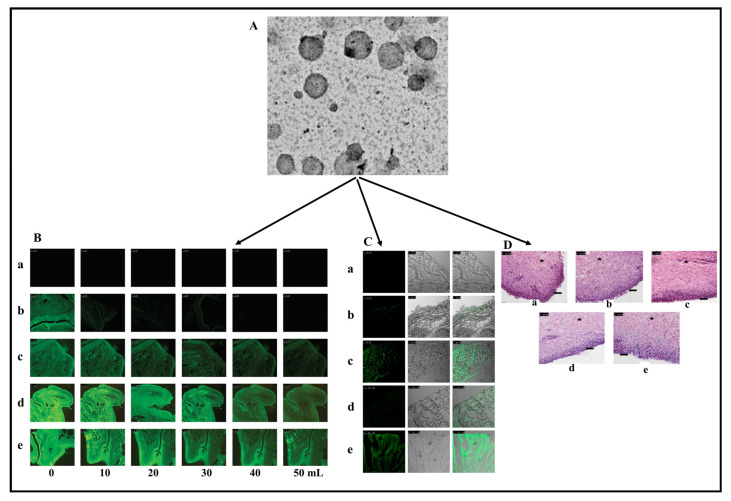
Trans emission electron microscopy of quercetin contained cationic solid lipid nanoparticles (**A**), fluorescence image indicating retentions of varied quercetin contained solid lipid nanoparticles preparation on bovine urinary bladder mucosa (**B**), confocal microscopic image indicating cross-section of bovine urinary bladder mucosa in presence of fluorescence dye (**C**), and microscopic image of bladder mucosa treated with quercetin-loaded cationic solid lipid nanoparticle formulations; represents Urothelium and lamina propria (**D**). Reproduced with permission from [79] under the Creative Commons attribution (CC BY 4.0) license.

**Figure 6 pharmaceuticals-17-00329-f006:**
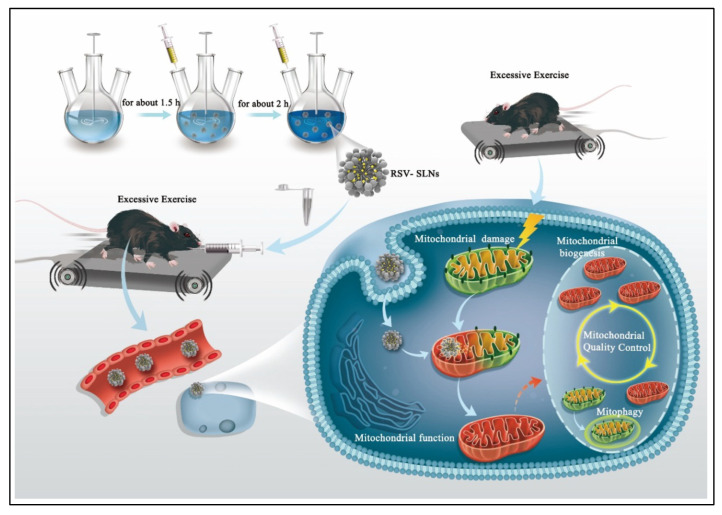
Resveratrol-incorporated solid lipid nanoparticles fabricated using emulsification techniques at a low-temperature solidification process for improved performance via modulating mitochondrial quality control. Reproduced with permission from [98] under the Creative Commons attribution (CC BY 4.0) license.

**Figure 7 pharmaceuticals-17-00329-f007:**
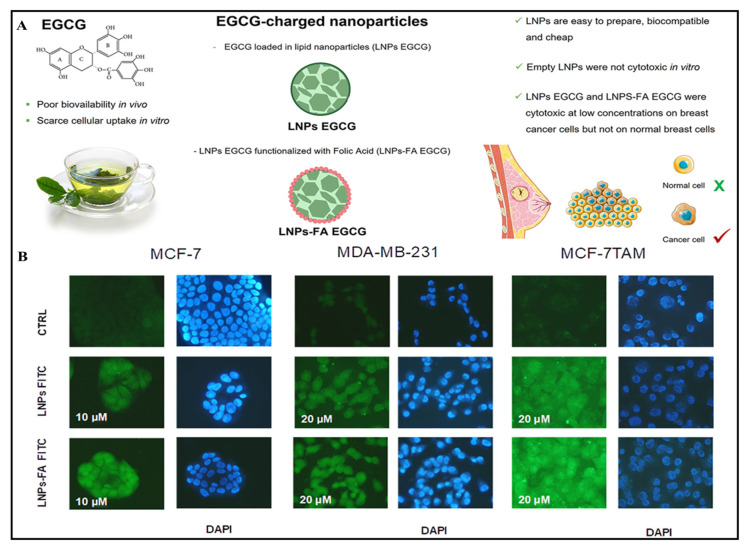
Systemic illustration indicating fabrication of epigallocatechin-3-gallate lipid nanoparticles with or without folic acids (**A**). Fluorescence image indicating cellular internalization of epigallocatechin-3-gallate lipid nanoparticles with or without folic acids labeled with 4,6-diamidino 2-phenylindole dihydrochloride (DAPI) fixed with 1% formalin (**B**). Moreover, these cellular indicators indicate cellular survival percentage with an indication of the toxic nature of treated cancer cells. Reprinted with permission from [104] under CC-BY-NC-ND 4.0.

**Figure 8 pharmaceuticals-17-00329-f008:**
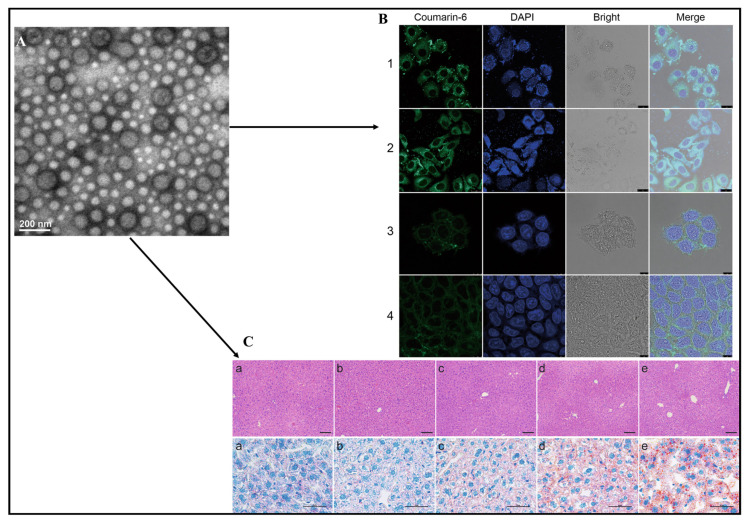
Transmission electron microscopy of chitosan-modified silymarin-incorporated lipid nanoparticles (**A**), cellular morphology after oil red O staining as indicated by confocal microscopy (**B**); were uptake of curcumin-loaded LPNs (C-LPNs) by HepG2 fatty liver cells treated with 1% medical fat emulsion (**B-1**); uptake of curcumin-loaded chitosan modified LPNs (CC-LPNs) by HepG2 fatty liver cells treated with 1% medical fat emulsion (**B-2**); uptake of C-LPNs by Caco-2 cells (**B-3**); uptake of CC-LPNs by Caco-2 cells (**B-4**), and Hematoxylin and eosin staining results for normal diet (**C-a**), chitosan-fortified silymarin lipid nanoparticles (**C-b**), alone silymarin lipid nanoparticles (**C-c**), silymarin suspension (**C-d**), high-fat diet (**C-e**) e as indicated on the upper portion of image under label C and oil red O staining results indicated by normal diet (**C-a**), CS-LPNs (**C-b**), S-LPNs (**C-c**), S-suspension (**C-d**), HFD (**C-e**) as indicated on the lower portion of image under label C. Reproduced with permission from [112] under the Creative Commons attribution (CC BY 4.0).

**Table 2 pharmaceuticals-17-00329-t002:** Recent report on the lipid-based nanotechnology in delivering bioactive compounds with improved therapeutic efficacy.

Patented Lipid-Based Nanoparticles Encapsulating Bioactive Compound	Site	Applications	Country of Patent	Reference
Floating pills contain SLNs loaded with an extract of the *Ficus benjamina* aerial part.	Oral	IN2960/DEL/2014	India	[139]
Increased oral bioavailability of *Tripterygium* glycoside due to increased dissolvability	Oral	CN200910034968	China	[140]
Nanoparticles fabricated using cacao butter can be used to administer medications parenterally.	Parenteral	WO2009102121	Korea	[141]
Delivery of nanoparticles improves the stability and effectiveness of UV filters.	Dermal	WO0103652	Canada	[142]

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
