# Peer review of "Lipid-Based Nanoparticles in Delivering Bioactive Compounds for Improving Therapeutic Efficacy"

_pharmaceuticals, 2024, doi:10.3390/ph17030329_

Round 1

Reviewer 1 Report

Comments and Suggestions for Authors

The manuscript requires several improvements for better clarity and comprehension:

 Avoid using abbreviations without spelling out the word first (e.g., GIT; EGCG).

The abstract should mention the main bioactive molecules and the types of lipid-based nanocarriers discussed in the review.

Modify keywords to terms that were not used in the title or abstract.

Include objectives at the end of the introduction section to better contextualize the review's contribution. Also, mention the main novelties, especially given other recent studies on the same subject (e.g., 10.1016/j.indcrop.2023.117602).

In Figure 1, include images representing the types of nanocarriers investigated in the review. Consider adding chemical structures of biocompounds to familiarize readers with the studied molecules and mention the extraction method as a restriction.

Consider including tables detailing the main properties of cited lipids (Section 3) and types of systems, molecules, preparation methods, compositions, and outcomes (Section 4) to facilitate objective reading.

Include a brief section discussing the main plant sources of the biomolecules described in the review.

Consider removing Figure 8 if it does not provide informative content.

Include a brief paragraph discussing the study's limitations to enhance data interpretation.

Comments on the Quality of English Language

Address typographical errors throughout the text.

Author Response

Thank you for your time and effort on reviewing our manuscript. We have responded the reviewer's comments point by point as shown in the attached file. Thank you again for your consideration.

Reviewer 2 Report

Comments and Suggestions for Authors

The review faces a topic of very actual and prospective interest for the scientific community. However, some concerns are raising to me from its evaluation:

i) in some points, the level of novelty of bibliographic information is poor. Some sections only reports publication of 20 years ago! Authors are invited to revise this aspect, updating when possible the literature or reducing/removing the discussion on compounds that have not been studied in recent times.

ii) I did not find any reference to NLC, that are the 'newer' version of SLN for many aspects and in more recent papers.

iii) the entire section 5 must the improved in quality and length, there are only short considerations on safety, patents etc., with respect to the other parts of the review.

Author Response

(The authors gave the same response as above.)

Reviewer 3 Report

Comments and Suggestions for Authors

This is an interesting review about Lipid-based Nanoparticles in Delivering Bioactive Compounds for Improving Therapeutic Efficacy. I suggest it for publication after the following minor points are well addressed.

1. Section 3.2 Synthetic lipids, one study (Langmuir 2019, 35, 18, 6048–6054) should be included in this section. 

2. The resolution of figure 3 is not enough.

3. One figure should be added to show the different chemical structures of the lipids mentioned in this review.

Comments on the Quality of English Language

Minor editing of English language required

Author Response

Thank you for your time and effort on reviewing our manuscript. We have responded the reviewer's comments point by point as shown in the attached file. Thank you very much for your consideration.

Reviewer 4 Report

Comments and Suggestions for Authors

The review entitled "Lipid-based Nanoparticles in Delivering Bioactive Compounds for Improving Therapeutic Efficacy" by Patel et al. is an interesting paper describing the different strategies used by researchers to encapsulate bioactive compounds in lipidic nanoparticles. The review is well organized, but I have few comments for the Authors:

1) In the abstract, I suggest to replace the acronym "GIT" with the corresponding words; it is not immediate for the reader to understand what the Authors are referring to.

2) on line 48: please replace "the therapeutic efficacy in clinical is..." with "the therapeutic efficacy in clinics is..."

3) Lines 64-66: I think this sentence is not well constructed; please re-write it in a more gramatically correct form.

4) On line 72, the Authors mention the different types of lipid-based nanoparticles used to encapsulate bioactive compounds: I think the Authors are forgetting to mention extracellular vesicles, exosomes and biomimetic vesicles, which have been attracted great deal of interest in recent years, also for deliverying bioactive principles. Therefore, the Authors should mention these nanosystems and they should also add a section dedicated to exosomes, extracellular vesicles and biomimetic vesicles.

 I suggest the Authors to mention the following paper, where the Authors can find references to numerous works dedicated to the topic:

 - Baldassari, S. et al. Phytochemicals and Cancer Treatment: Cell-Derived and Biomimetic Vesicles as Promising Carriers. Pharmaceutics 2023, 15, 1445. https://doi.org/10.3390/pharmaceutics15051445

5) Please at the end of the introduction add some lines explaining how the review will be organized, and how the topics have been divided.

6) Figure 1: I do not really understand the point of this image. What does it means the pentagon entitled “Problem and Solution”? I do not think that a flow diagram is the best type of representation to illustrate the content the Authors want to express.

7) Lines 104-109: the Authors should add some lines explaining what is the advantage of SLNs compared to micelles

8) line 162: please replace “negative charge” with “negatively charged”

9) Line 257: I do not know the meaning of the word “plaas”. Checked on different online translators, it is not English language. Please, remove this word and add the appropriate English word.

Comments on the Quality of English Language

English language is not very fluent, but I think it is acceptable. I pointed out some sentences that should be modified, because they are not clear.

Author Response

(The authors gave the same response as above.)

Round 2

Reviewer 1 Report

Comments and Suggestions for Authors

The authors made significant improvements in the document. The revised version is better than the previous one, but I still need to mention that the resolution of the included images is not optimal. The chemical structures must be revised before the publication of this manuscript. 

Author Response

Thank you for your time and efforts for reviewing our manuscript. 

Reviewer 2 Report

Comments and Suggestions for Authors

Suggestions for revision have been well addressed in the revised version of the ms. Quality improvement is now acceptable.

Author Response

Thank you for your valuable and constructive comments. We appreciate your time and efforts reviewing our manuscript.

Reviewer 4 Report

Comments and Suggestions for Authors

I think the Authors answered to almost all the points. However, they decided to ignore my suggestion to cite the following review: 

Baldassari, S. et al. Phytochemicals and Cancer Treatment: Cell-Derived and Biomimetic Vesicles as Promising Carriers. Pharmaceutics 202315, 1445. https://doi.org/10.3390/pharmaceutics15051445

They cited another review, focusing on the definition of exosomes and on the different strategies to isolate exosomes. I think it is ok to cite this work, but I still think that it would be important to cite the very recent (2023) review by Baldassari et al., since this review focuses on the use of exosomes to encapsulate bioactive compounds, thus perfectly fitting the topic of the present manuscript.

Author Response

We appreciate your suggestion. The reference “Baldassari, S.; Balboni, A.; Drava, G.; Donghia, D.; Canepa, P.; Ailuno, G.; Caviglioli, G. Phytochemicals and Cancer Treatment: Cell-Derived and Biomimetic Vesicles as Promising Carriers. Pharmaceutics 2023, 15, 1445” was cited as reference #15 of the revised manuscript. Thank you very much.